# Development of a Gait Analysis Application for Assessing Upper and Lower Limb Movements to Detect Pathological Gait

**DOI:** 10.3390/s24196329

**Published:** 2024-09-30

**Authors:** Atsuhito Taishaku, Shigeki Yamada, Chifumi Iseki, Yukihiko Aoyagi, Shigeo Ueda, Toshiyuki Kondo, Yoshiyuki Kobayashi, Kento Sahashi, Yoko Shimizu, Tomoyasu Yamanaka, Motoki Tanikawa, Yasuyuki Ohta, Mitsuhito Mase

**Affiliations:** 1Department of Neurosurgery, Nagoya City University Graduate School of Medical Science, Nagoya 467-8601, Japan or taishaku@med.nagoya-cu.ac.jp (A.T.); mtnkw@med.nagoya-cu.ac.jp (M.T.); mitmase@med.nagoya-cu.ac.jp (M.M.); 2Interfaculty Initiative in Information Studies, Institute of Industrial Science, The University of Tokyo, Tokyo 113-8654, Japan; 3Department of Behavioral Neurology and Cognitive Neuroscience, Tohoku University Graduate School of Medicine, Sendai 980-8575, Japan; chi.iseki@gmail.com; 4Division of Neurology and Clinical Neuroscience, Department of Internal Medicine III, Yamagata University School of Medicine, Yamagata 990-9585, Japan; toshikon@med.id.yamagata-u.ac.jp (T.K.); yasuyuki@med.id.yamagata-u.ac.jp (Y.O.); 5Digital Standard Co., Ltd., Osaka 530-0014, Japan; y.aoyagi@digital-standard.com; 6Shin-Aikai Spine Center, Katano Hospital, Katano 576-0043, Japan; uedashigeo@yahoo.co.jp; 7Human Augmentation Research Center, National Institute of Advanced Industrial Science and Technology (AIST), University of Tokyo, Kashiwa II Campus, Chiba 277-0882, Japan; kobayashi-yoshiyuki@aist.go.jp; 8Department of Rehabilitation, Nagoya City University Graduate School of Medical Science, Nagoya 467-8601, Japan; ptkento1204@live.jp (K.S.); otyokos@med.nagoya-cu.ac.jp (Y.S.)

**Keywords:** deep learning, motion tracking, markerless motion capture, quantitative gait assessment, smartphone device, idiopathic normal-pressure hydrocephalus, Hakim’s disease, Parkinson’s disease, cervical myelopathy

## Abstract

Pathological gait in patients with Hakim’s disease (HD, synonymous with idiopathic normal-pressure hydrocephalus; iNPH), Parkinson’s disease (PD), and cervical myelopathy (CM) has been subjectively evaluated in this study. We quantified the characteristics of upper and lower limb movements in patients with pathological gait. We analyzed 1491 measurements of 1 m diameter circular walking from 122, 12, and 93 patients with HD, PD, and CM, respectively, and 200 healthy volunteers using the Three-Dimensional Pose Tracker for Gait Test. Upper and lower limb movements of 2D coordinates projected onto body axis sections were derived from estimated 3D relative coordinates. The hip and knee joint angle ranges on the sagittal plane were significantly smaller in the following order: healthy > CM > PD > HD, whereas the shoulder and elbow joint angle ranges were significantly smaller, as follows: healthy > CM > HD > PD. The outward shift of the leg on the axial plane was significantly greater, as follows: healthy < CM < PD < HD, whereas the outward shift of the upper limb followed the order of healthy > CM > HD > PD. The strongest correlation between the upper and lower limb movements was identified in the angle ranges of the hip and elbow joints on the sagittal plane. The lower and upper limb movements during circular walking were correlated. Patients with HD and PD exhibited reduced back-and-forth swings of the upper and lower limbs.

## 1. Introduction

Pathological gait is a deviation from normal gait, such as shuffling, short-stepped, freezing, wide-based, festination, hemiplegic, spastic, and ataxic gaits [1,2,3,4,5,6]. However, these pathological gait features are subjectively evaluated, without any standardized rating systems. Qualitative or semiquantitative subjective assessments, including medical record input, are simple and widely used in clinical practice; however, their validity and reliability are generally low [1,2]. The characteristics of pathological gait are described as reduced stride length and diminished step height in patients with Parkinson’s disease (PD) and Hakim’s disease (HD) (synonymous with idiopathic normal-pressure hydrocephalus; iNPH) [6,7,8,9,10,11,12], asymmetry of step length and leg swing velocity in patients with hemiplegia [13,14], and variation of stride length and step width in patients with ataxia [15]. The optical three-dimensional (3D) motion capture technology using multipoint cameras and numerous body markers enables the analysis of the spatiotemporal kinetics [16,17,18,19]. However, it requires excessive time, effort, technical support, and cost. Therefore, a number of markerless motion capture systems, based on 3D optical sensing technology and deep learning, have recently been reported [20,21,22,23,24,25,26,27,28]. We developed a novel iOS application, available in daily clinics, to analyze gait more convenient, i.e., the Three-Dimensional Pose Tracker “https://digital-standard.com/tdpt-sdk_lp/en/index.html (accessed on 15 August 2024)” for Gait Test (TDPT-GT), which generates the 3D movement information for the entire body using video images captured by a single iPhone camera [29]. In addition, we built additional software for 2D projection to the sagittal, coronal, and axial planes relative to the body axis from 3D coordinates generated by the TDPT-GT to analyze the angles of the joints. We found that the characteristics of the legs and feet of the person with a shuffling gait exhibited a hip joint movement of less than 30° and a vertical amplitude of the heel of less than 0.1 on the sagittal projection [30]. Gait is not only constructed by the legs; arm swings are also an essential element in regulating the whole-body dynamic stability and contribute to reducing the kinetic energy expenditure [31,32,33,34]. In PD patients, decreased arm swing and postural instability are reported, in addition to their gait disturbance [19,35,36,37]. However, in patients with HD, their upper limb movements were not mentioned as pathologic in the past because typical symptoms of HD were identified as “lower body parkinsonism” [38,39]. Cervical myelopathy (CM) shows spastic gait disturbance, which is also diverse, with symptoms affecting the upper limbs, to some extent [40,41]; this disturbance is sometimes silent, or difficult to understand as gait disturbance.

Gait disturbance can be analyzed using our TDPT. However, differential diagnosis between the diseases is still challenging. In the present study, we aim to identify the gait characteristics, including whole body information, related to HD, PD, and CM, specifically revealing the relationship between the motion of the upper and lower limbs.

## 2. Materials and Methods

### 2.1. Ethical Approvals

The study design and protocol of this study were approved by the Ethics Committee for Human Research of Nagoya City University Graduate School of Medical Science (IRB number: 60-22-0111), Shiga University of Medical Science (R2019-337), and Yamagata University School of Medicine (protocol code: 2021-10). Written informed consent was provided by all volunteers and patients.

### 2.2. Study Population

The details of the study population have been described in our previous publications [29,30,42]. In brief, the inclusion criteria for all participants, including patients with gait disturbance, were as follows: independence in their ability to walk without assistance and the ability to make independent decisions. The dataset contained healthy volunteers, aged ≥ 20 years, and patients diagnosed with HD, PD, and CM from the four collaborating institutions. One cohort included volunteers aged ≥ 60 years who participated in a Takahata population-based cohort study [43], and the other dataset comprised healthy volunteers aged ≥ 20 years, recruited by open recruitment from 2020 to 2022. HD was diagnosed based on the third edition of the Japanese guidelines for the management of iNPH [4], and PD was diagnosed based on the clinical criteria proposed by the Movement Disorder Society [44]. The gaits of the patients with HD, PD, and CM were evaluated using the TDPT-GT while simply walking around a 1 m diameter circle twice at several timings; for example, before and after a cerebrospinal fluid (CSF) tap test or shunt surgery for patients with HD, before and after medication intake (dopamine test) for those with PD, and before and after spine surgery for those with CM. At each timepoint, measurements were obtained twice. The changes and degree of improvement in symptoms before and after these interventions varied significantly among individuals and were not taken into consideration.

### 2.3. Three-Dimensional Pose Tracker for Gait Test (TDPT-GT) App for 3D Human Motion Estimation

This TDPT-GT app was designed to process three consecutive image captures instead of static images, utilizing a modified ResNet34 model trained with the Adam optimizer. To create the original training dataset for 3D pose tracking, we utilized over 100 humanoid CG characters in VRM format, each annotated with 3D coordinates for 24 key points, e.g., the nose, ears, and joints. The coordinates were relative to the character’s body center and standardized for the distance from the body center to the head and ankles. Using the Unity system and a virtual camera, we captured more than one million 2D images, consisting of three consecutive frames at up to 60 frames per second. These 448 × 448 pixel RGB images featured the characters performing walking and dancing movements, which then served as training data for the deep learning process. The training objective involved minimizing L2 loss for four outputs related to 3D coordinates. To estimate human motion, a 3D heatmap method was employed, dividing space into 28 × 28 × 28 blocks across 24 key points. The model identified the most probable block for each key point and matched it with key points from the previous two frames for tracking motion. Fine-tuning of the X, Y, and Z coordinates within each block was performed using the hyperbolic tangent (tanh) function. Finally, features from the convolution layers were passed through fully connected layers, outputting normalized probabilities for multiclass classification. This method enabled accurate 3D motion estimation, enhancing its use for motion tracking and analysis (Appendix A).

### 2.4. Projection of Relative Coordinates on the Body Axis Planes

The 3D relative coordinates of the 24 key points were estimated using the TDPT-GT while simply walking around a 1 m diameter circle twice. For each key point, the 3D coordinates observed were raw, and those smoothed by applying a low-pass filter were automatically saved at 30 fps in the iPhone SE2 in a csv file format. From these 3D coordinates, a confidence score of ≥0.7 represents the accuracy and certainty of the 3D coordinates estimated by deep learning. A confidence score closer to 1 indicated higher certainty, while scores below 0.7 were considered unreliable and were excluded from further calculations of the 2D relative coordinates projected onto the tri-axis planes. As shown in Figure 1, the normal vector to a plane passing through the navel center and right and left shoulder points was defined as $\underset{UF}{\to }$, and the normal vector to a plane passing through the navel center and right and left hip joints was defined as $\underset{DF}{\to }$. The $\underset{f}{\to }$ for the forward direction of the body axis was combined with the $\underset{UF}{\to }$ and $\underset{DF}{\to }$. To estimate the 2D relative coordinates of the legs, the perpendicular vector $\underset{c}{\to }$ to the vector $\underset{f}{\to }$ was set from the navel center to the center between the bilateral hip joints, whereas the perpendicular vector $\underset{cc}{\to }$ to the vector $\underset{f}{\to }$ was set from the navel center to the center between the bilateral shoulder joints for the upper limbs. The $\underset{n}{\to }$ was set as orthogonal to the two vectors of $\underset{f}{\to }$ and $\underset{c}{\to }$ or $\underset{cc}{\to }$. The relationships among these three vectors were $\underset{f}{\to }$·$\underset{c}{\to }$ = $\underset{f}{\to }$·$\underset{n}{\to }$ = $\underset{n}{\to }$·$\underset{c}{\to }$ = 0 or $\underset{f}{\to }$·$\underset{cc}{\to }$ = $\underset{f}{\to }$·$\underset{n}{\to }$ = $\underset{n}{\to }$·$\underset{cc}{\to }$ = 0. The sagittal, coronal, and axial planes for the upper and lower limbs were defined as planes composed of $\underset{f}{\to }$ and $\underset{c}{\to }$, $\underset{f}{\to }$ and $\underset{n}{\to }$, and $\underset{n}{\to }$ and $\underset{c}{\to }$, respectively. Each 2D coordinate projected on the sagittal, coronal, and axial planes of each 3D relative coordinate was the intersection of the projected plane and the normal line relative to the projected plane. Furthermore, the 2D relative coordinates of the bilateral hip joints, knees, heels, and toes were corrected using the total leg length as 1 on the projected plane, and those of the upper body points were corrected using the total length of the upper body from the navel center to the head center as 1.

### 2.5. Data Processing for Gait Analysis

For each key point, 75% tolerance ellipses were drawn on the 2D plots projected onto the sagittal, coronal, and axial planes. The calculations included the area, center of the coordinates, tilt angles of the major and minor axes, and the lengths of the x- and y-axes of these ellipses. The area of each ellipse was computed using the following formula: length of the major axis × length of the minor axis × π. In the sagittal and coronal planes (Figure 2), the angle range of the hip joint was determined by the angle between two vectors, which was drawn from the center of the 75% tolerance ellipse for the hip joint to two points on the major axis of the knee ellipse. Similarly, each angle range was defined as the vector angle from the proximal joint to the distal joints.

The axial projection planes are shown as each part of the upper and lower limbs (Figure 3). Leg outward shifts are defined as the sum of the toe and heel outward shifts, which are associated with a wide-based gait [30]. Toe outward shifts are lateral deviations of the right and left x-coordinates of the toe ellipse centers from those of the heel ellipse centers. Heel outward shifts are lateral deviations of the right and left x-coordinates of the heel ellipse centers from those of the hip ellipse centers (Figure 3, upper). Arm outward shifts are the sums of the hand and elbow outward shifts. Hand outward shifts are lateral deviations of the right and left x-coordinates of the hand ellipse centers from those of the elbow ellipse centers. Elbow outward shifts are lateral deviations of the right and left x-coordinates of the elbow ellipse centers from those of the shoulder ellipse centers (Figure 3, lower).

### 2.6. Statistical Analysis

The Pearson’s correlation coefficient (r) and 95% confidential intervals (CIs) were used for the analysis of the distributions and correlations of the aforementioned parameters of the upper and lower limbs. Statistical significance was assumed at a probability (*p*) value of <0.05. All missing data points were treated as deficit data that did not affect other variables. Statistical analyses were performed using R (version 4.4.0; the R Foundation for Statistical Computing; “http://www.R-project.org (accessed on 15 August 2024) ”

## 3. Results

### 3.1. Clinical Characteristics

In total, 427 participants, including 122 patients with HD, 12 patients with PD, 93 patients with CM, and 200 healthy volunteers, underwent 1491 TDPT-GT measurements (with an average of 3.5 measurements per person). Table 1 summarizes the participants’ characteristics. For 51 healthy volunteers participating from one collaborating institution, sex and age were not recorded; however, all were over 20 years old.

The angle ranges of the hip and knee joints on the sagittal projection plane were significantly smaller in the following order: healthy volunteers > CM > PD > HD patients (Figure 4, upper right). In contrast, the angle ranges of the shoulder and elbow joints were significantly smaller in the following order: healthy volunteers > CM > HD > PD patients (Figure 4, upper left). Ellipse areas of the elbows, hands, knees, and heels on the sagittal plane were also compared among the four groups. The heel ellipse area was significantly smaller in the following order: healthy volunteers > CM > PD > HD patients, which was similar to the angle range of the knee joint (Figure 4, lower right). However, the elbow ellipse area was significantly larger in patients with HD and PD than in patients with CM and healthy volunteers (Figure 4, lower left). In particular, patients with HD and PD exhibited a smaller angle range of shoulder joint motion in the anterior–posterior direction than did healthy volunteers; however, a larger elbow elliptical area was observed among them. This indicates decreased arm swing but increased vertical movement of the elbow joint in patients with HD and PD.

Significant differences in the angle ranges and ellipse areas of the upper and lower limb joints on the coronal projection plane were observed among the four groups, which are similar to those noted on the sagittal projection plane; however, the differences were all smaller, and the characteristics were more difficult to capture than those on the sagittal projection plane (Figure 5).

The outward shift of the leg, heel, and toe on the axial projection plane was significantly greater in the following order: healthy volunteers < CM < PD < HD patients (Figure 6, lower). In particular, the leg outward shift in patients with HD was significantly greater, averaging >15% of the leg length, whereas in healthy volunteers, it showed a negative value, indicating an inward shift (Figure 6, lower left). The outward shift of the upper limb, elbow, and hand on the axial projection plane was significantly smaller in the following order: healthy volunteers > CM > HD > PD patients (Figure 6, upper). In particular, the arm outward shift in patients with PD was significantly smaller, averaging <100% (Figure 6, upper left).

### 3.2. Indices of 2D Relative Coordinates on the Tri-Axial Projection Planes

The parameters of the angle ranges of the joints and the 75% tolerance ellipse between the upper and lower limbs of the 2D relative coordinates projected onto the sagittal, coronal, and axial planes were comprehensively compared. The highest correlation coefficient (*r*) between the upper and lower limb movements in the 1491 TDPT-GT measurements was 0.44 (95% CI: 0.40–0.48) between the angle ranges of the hip and elbow joints on the sagittal projection plane (Figure 7, left). Among the 75% tolerance ellipse areas of each joint in the upper and lower limbs, the *r* between the elbow and hip joints was the highest at 0.34 (95% CI: 0.30–0.38) (Figure 7, left). In the correlation analysis by subgroup, the *r* between the angle ranges of the hip and elbow joints in patients with PD was the highest at 0.47 (95% CI: 0.27–0.64) (Table 2).

For the angle ranges and ellipse areas of the joints in the upper and lower limbs on the right and left sides, no stronger correlations were observed than the left–right averages on the sagittal projection planes. Similarly, for the parameters on the coronal and axial projection planes, no stronger correlations were observed than those on the sagittal projection planes.

## 4. Discussion

We developed a simple 3D motion capture system using a smartphone, and 2D coordinates projected onto the three body-axis planes can objectively and quantitatively measure the upper and lower limb movements of various pathological gait patterns in patients with neurological diseases, including HD (iNPH), PD, and CM. Several previous studies have compared the pathological gaits of patients with HD and PD [6,45,46]. The common gait features between HD and PD are shuffling, short-stepping, freezing, and festination, whereas the most distinguishable gait feature is a wide-based gait, specific to HD, and a closed gait, specific to PD [6]. Furthermore, the most characteristic gait feature specific to CM is a spastic gait. In this study, we demonstrated that patients with HD exhibited significantly narrower angle ranges of the hip and knee joints on the sagittal projection plane than did patients with PD. This means that patients with HD display a more severe shuffling and short-stepped gait than do patients with PD. Furthermore, patients with HD exhibited a significantly wider outward shift of the legs on the axial projection plane than did the other patient groups and healthy volunteers. This means that patients with HD exhibited a more severe wide-based gait. However, patients with PD also displayed a significantly wider outward shift than did patients with CM and healthy volunteers. As mentioned in our previous study [30], a possible explanation is that our assessment task involves walking around a circle with a diameter of 1 m, which is different from a straight walking task. Therefore, patients with HD and PD exhibiting balance impairments would find it easier to open their legs when walking in a circle. However, as for the upper limbs, patients with PD showed less arm swing, not only regarding back and forth motion on the sagittal plane but also for the outward shift on the axial plane, particularly concerning hand movement to the elbow joint on the axial plane, than did the other patient groups. This is known as a typical upper limb movement disorder in PD due to stiffness and bradykinesia [44], which we were able to quantify in this study.

Although arm swing during gait is significantly associated with regulating whole-body dynamic stability and contributes to reducing the kinetic energy expenditure [31,32,33,34], the upper body, including arm swing, has not often been the subject of extensive focus in gait analysis. The typical feature of arm swing during the gait cycle in PD includes reduced amplitude and speed, asymmetry, and asynchrony of movement, which occur more frequently with progression and increasing disease severity [35,36,37]. Although the characteristics of gait disturbance in patients with HD affect their lower body movements [6,38], significant improvements in upper limb movements have been reported in patients with HD after the CSF tap test and shunt surgery [39,47,48,49]. Few studies have compared upper limb movements during gait in these diseases, despite the significant role of the upper limb in bipedal walking. A recent study using the MyoMotion system (Noraxon USA Inc., Scottsdale, AZ, USA), which requires setting up 16 inertial measurement units with a three-axis accelerometer, three-axis gyroscope, and three-axis magnetometer on the entire body, reported that the ranges of motion of the hip and knee joints were significantly decreased in patients with HD compared with those in patients with CM [50]. Using the comprehensive assessment of the relationships between the motion parameters of the upper and lower limbs, we found a significantly high correlation between the angle ranges of the hip and elbow joints on the sagittal projection plane. This relationship was the strongest in PD patients but weaker in those with HD. In our previous study using random fluctuation analysis [46], HD patients exhibited significantly more severe pathological fluctuations in both upper and lower limb movements compared to PD patients and healthy volunteers. The results of this study further suggest that HD patients exhibit poorer coordination between upper and lower limb movements compared to PD patients.

Our TDPT system and an additional motion analysis method could easily evaluate not only leg motions but also pathological arm swings during walking in a short amount of time, in a small space, and using minimal equipment. This smartphone-based sensor allows gait analysis to be brought into daily clinics by employing AI motion capture. In the future, big data from multicenter collaborative studies obtained using this application will be able to score the severity of pathological gaits and detect slight changes in gait disturbances more accurately. Then, the analyzed results can be provided to participants promptly to encourage them to exercise, rehabilitate, or live healthier.

This study has some limitations that warrant discussion. First, the patients with HD, PD, and CM were subjectively evaluated for the presence or absence of shuffling, short-stepped, wide-based, freezing, and spastic gaits and instability following classical experimental diagnosis. Subjective evaluation remains ambiguous and often varies, with judgments differing among experts [2]; however, we cannot avoid this subjectivity when observing patients and developing an objective evaluation. Second, as a control group, this study used healthy volunteers who were significantly younger than the HD, PD, and CM groups. Older people tend to walk slowly, with shorter steps and wider stances due to decreased forward propulsion, etc., when compared with the gaits of young people [51,52,53]. Age-matched controls are better for comparing gait features. Third, the 2D relative coordinates projected onto the sagittal, coronal, and axial planes of the body axis, based on the navel as the center without the ground information, could not be compared with those used in previous conventional gait research measured using the distance based on the floor surface. However, the indices of the present study, i.e., the gait cycle time, stride length, step frequency, and angle range of the joint, can be compared with those in previous research. Finally, the reliability and validity of the 3D relative coordinates estimated by the TDPT-GT application have not yet been fully verified in this study, although gait analysis using monocular camera deep learning-based markerless motion capture has been extensively studied in recent years [24,25,26,27,54,55].

## 5. Conclusions

This study shows that upper limb motions can be evaluated, in addition to the previously reported evaluation of lower limb motions, using 2D coordinates projected onto the sagittal, coronal, and axial planes relative to the body axis from the 3D relative coordinates estimated using the AI-based smartphone application TDPT-GT. TDPT-GT allows for the quantitative differentiation of pathological gaits in patients with various neurological gait disorders. During the gait cycle, the upper and lower limbs move in coordination, and in pathological gait, in addition to leg movements, upper limb movements are also impaired. Patients with HD exhibited significantly narrower angle ranges of the hip and knee joints on the sagittal plane and a markedly outward shift of the legs on the axial plane, e.g., a shuffling, short-stepped, and wide-based gait, whereas patients with PD displayed less back and forth and less outward arm swing motion than patients with HD or CM. This is known as a typical upper limb movement disorder in PD. This AI-based application quantifies pathological gaits using both upper and lower limb motions, making it possible for not just doctors and researchers but also patient families to easily detect fall risks and gait disorders at an early stage, which will help prevent the progression of conditions that would require nursing care.

## Figures and Tables

**Figure 1 sensors-24-06329-f001:**
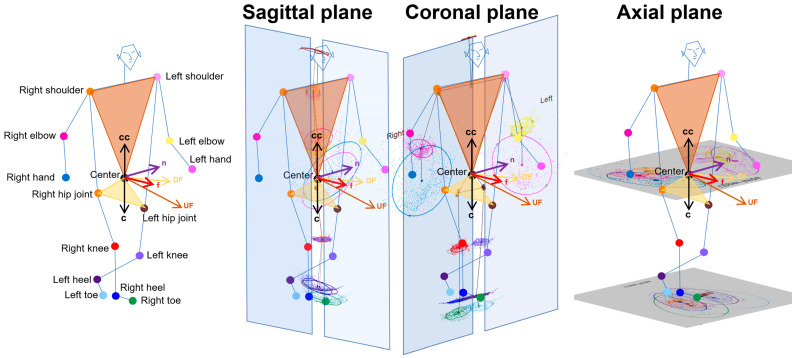
Two-dimensional relative coordinates projected onto the sagittal, coronal, and axial planes during normal walking of a healthy volunteer.. The forward direction vector relative to the body axis (f: red arrow) was obtained from the composite vector of the normal vector to the plane (ivory) formed by the three points of the navel and both hip joints (DF: ivory arrow) and the normal vector to the plane (brown) formed by the three points of the navel and both shoulder joints (UF: brown arrow). Next, the vector directed from the navel to both hip joints (c: black arrow) and the vector directed from the navel to both shoulder joints (cc: black arrow), as well as the vector orthogonal to these and the forward direction vector f (n: purple arrow), were determined. As a result, three axes and three planes (sagittal, coronal, and axial planes) for the upper and lower body were formed.

**Figure 2 sensors-24-06329-f002:**
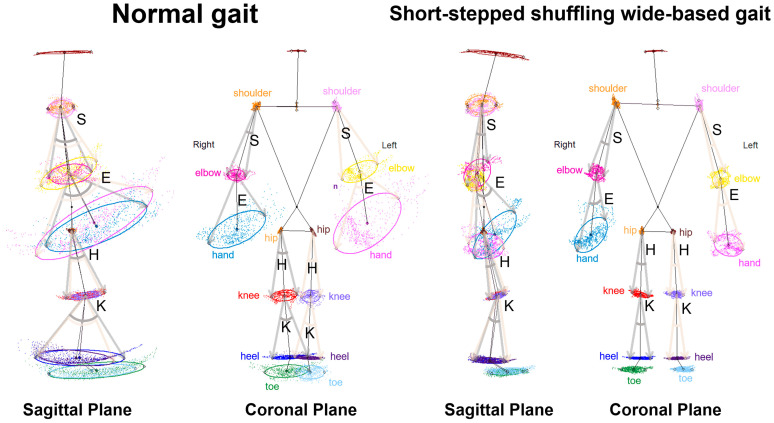
The movement of body points projected onto the sagittal and coronal planes. The motions of a healthy volunteer performing a normal gait (**left**) and a short-stepped shuffling wide-based gait (**right**). Colored coordinates and 75% tolerance ellipses of all plots of the body points were shown on the sagittal and coronal projection planes. S: the angle range of the shoulder joint; E: the angle range of the elbow joint; H: the angle range of the hip joint; and K: the angle range of the knee joint.

**Figure 3 sensors-24-06329-f003:**
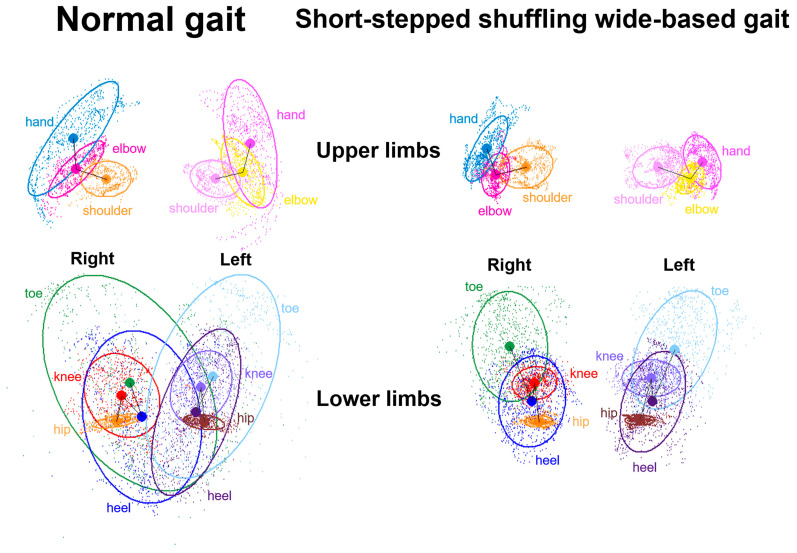
Two-dimensional relative coordinates projected onto the axial plane of the body axis. A representative healthy volunteer performed a normal gait (**left**) and a short-stepped shuffling wide-based gait (**right**) while the TDPT-GT was performed. Upper diagrams show chronological changes of the upper limbs, and lower diagrams show chronological changes of the lower limbs. The colored coordinates and 75% tolerance ellipses of all plots for the right and left shoulders, elbow, hand, hip joint, knee, right heel, and toe were shown on the axial projection plane.

**Figure 4 sensors-24-06329-f004:**
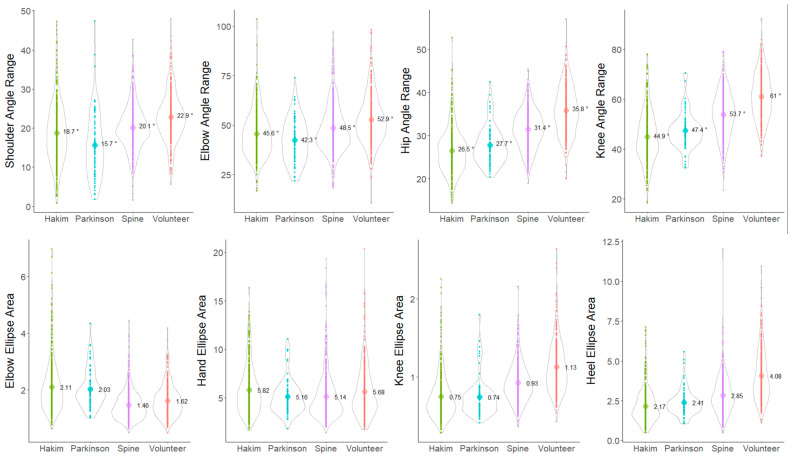
Distribution of the angle ranges of the joints and ellipse areas on the sagittal plane in the Hakim’s disease, Parkinson’s disease, cervical myelopathy, and healthy volunteer groups. Violin plots show the distribution in each group. The numbers in the violin plots show the averages of the angle ranges and the ellipse areas on the sagittal projection planes. Yellow-green indicates patients with Hakim’s disease, sky blue indicates patients with Parkinson’s disease, purple indicates patients with cervical myelopathy, and orange indicates healthy volunteers.

**Figure 5 sensors-24-06329-f005:**
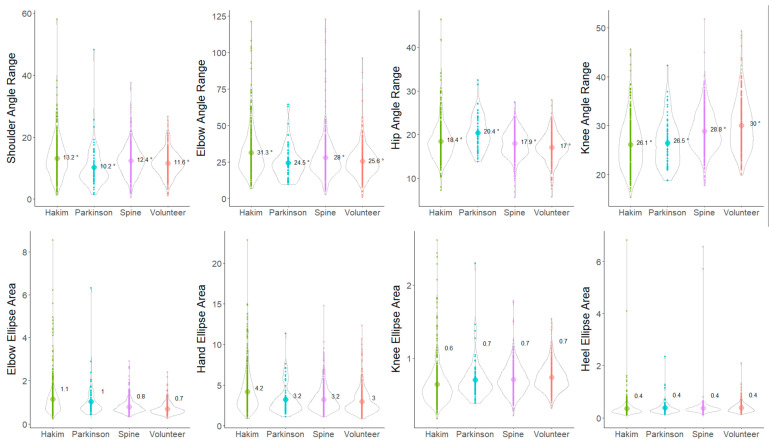
Distribution of the angle ranges of the joints and ellipse areas on the coronal plane in the Hakim’s disease, Parkinson’s disease, cervical myelopathy, and healthy volunteer groups. Violin plots show the distribution in each group. The numbers in the violin plots indicate the averages of the angle ranges and the ellipse areas on the coronal projection planes. Yellow-green indicates patients with Hakim’s disease, sky blue indicates patients with Parkinson’s disease, purple indicates patients with cervical myelopathy, and orange indicates healthy volunteers.

**Figure 6 sensors-24-06329-f006:**
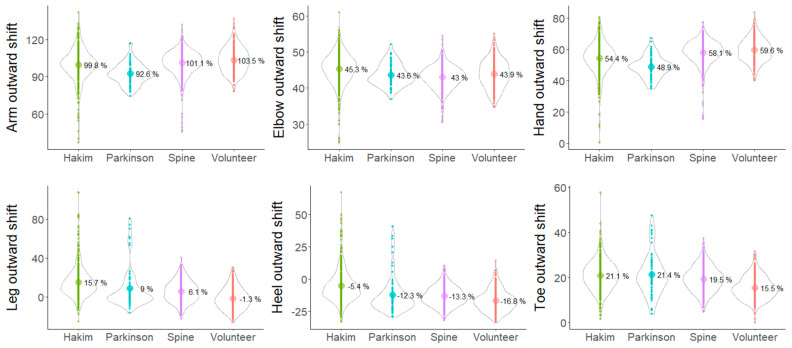
Distribution of the outward shift of the arms (elbows and hands) and legs (heels and toes) on the axial plane in the Hakim’s disease, Parkinson’s disease, cervical myelopathy, and healthy volunteer groups. Violin plots show the distribution in each group. The numbers in the violin plots indicate the averages of the parameters on the axial projection plane. Yellow-green indicates patients with Hakim’s disease, sky blue indicates patients with Parkinson’s disease, purple indicates patients with cervical myelopathy, and orange indicates healthy volunteers.

**Figure 7 sensors-24-06329-f007:**
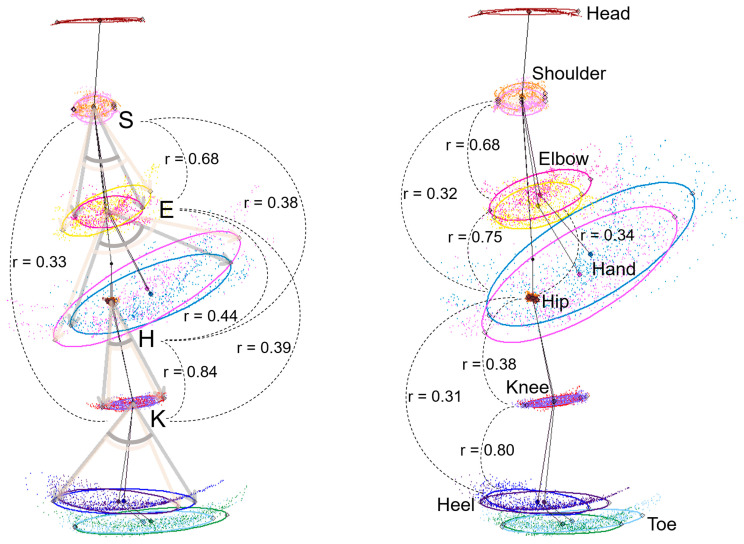
Correlation coefficients of the upper and lower limb parameters on the sagittal plane. The **left** figure shows the correlation between the angle ranges of the shoulder (S), elbow (E), hip (H), and knee (K) joints on the sagittal plane, and the **right** figure shows the correlations between the ellipse areas of the shoulders, elbows, hands, hip joints, knees, and heels on the sagittal plane. The numbers shown within the figure are the Pearson’s correlation coefficients (*r*).

**Table 1 sensors-24-06329-t001:** Clinical characteristics for subjects with each disease and healthy volunteers.

	Hakim’s	Parkinson’s	Cervical Myelopathy	Healthy Control
Total number	122	12	93	200
Sex (Male/female/unknown)	69:53:0	8:4:0	67:26:0	71:78:51
Mean ± SD of age (years)	75.9 ± 7.4	70.8 ± 7.9	66.1 ± 11.8	60.7 ± 20.2
Range of age (years)	60–93	56–83	42–88	20–91

SD: standard deviation.

**Table 2 sensors-24-06329-t002:** Pearson’s correlation coefficients (*r*) between the upper and lower limb movement parameters for subjects with each disease and healthy volunteers. (Bold indicates *r* > 0.3.)

1st Parameter	2nd Parameter	Hakim’s	Parkinson’s	Cervical Myelopathy	Healthy Control
Shoulder angle	Hip angle	0.23	**0.38**	**0.37**	**0.33**
Shoulder angle	Knee angle	0.16	0.27	**0.36**	0.28
Elbow angle	Hip angle	0.25	**0.42**	**0.38**	**0.32**
Elbow angle	Knee angle	**0.36**	**0.47**	**0.41**	**0.39**
Shoulder ellipse	Hip ellipse	**0.40**	0.03	0.17	**0.32**
Shoulder ellipse	Knee ellipse	0.15	−0.07	0.06	0.14
Shoulder ellipse	Heel ellipse	0.13	0.00	0.27	0.14
Elbow ellipse	Hip ellipse	**0.38**	0.28	0.25	**0.42**
Elbow ellipse	Knee ellipse	0.13	**0.35**	0.20	**0.31**
Elbow ellipse	Heel ellipse	0.08	**0.31**	**0.36**	**0.39**
Hand ellipse	Hip ellipse	0.25	0.29	0.28	**0.36**
Hand ellipse	Knee ellipse	0.21	**0.37**	0.25	**0.30**
Hand ellipse	Heel ellipse	0.24	**0.34**	**0.35**	**0.36**

## Data Availability

Data generated or analyzed during the study are available from the corresponding author upon reasonable request.

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
