# Peer review of "Development of a Gait Analysis Application for Assessing Upper and Lower Limb Movements to Detect Pathological Gait"

_sensors, 2024, doi:10.3390/s24196329_

Round 1

Reviewer 1 Report

Comments and Suggestions for Authors

General comments:

This study presents an interesting approach to analyzing human motion during gait by projecting upper and lower limb movement into 2D space and modeling the movement of each joint with elliptical drawings to compute quantitative geometric features for joint movements. The experiments were conducted with enough subjects including patients and health controls. The results revealed significant alterations of joint movement in patients compared to healthy control and explained how motions for related joints are correlated.

This manuscript is a well-written paper overall. However, providing more detailed descriptions would be beneficial for the better comprehension of readers.

Specific comments:

1. The descriptions of data acquisition are vague. Some explanations about TDPT-GT and data acquisition protocol may be found in section 1 and section 2.2. But it would be better to organize the description separately with a full explanation about it.

2. The 3D coordinate data with a confidence score of > 0.7 were used for analysis. What is the definition of confidence score and how they were calculated? Similarly, why the 75% tolerance ellipse were used to describe the distribution?

3. (Recommendation) The results showed general alternations of joint movements in patients compared to healthy control as group analysis. I think those features would have some relationship with quantitative gait indices or severity of mobility. How about performing subject-wise analysis to investigate those relationships?

Comments on the Quality of English Language

The quality of the written English Language is fine.

Author Response

General comments: This study presents an interesting approach to analyzing human motion during gait by projecting upper and lower limb movement into 2D space and modeling the movement of each joint with elliptical drawings to compute quantitative geometric features for joint movements. The experiments were conducted with enough subjects including patients and health controls. The results revealed significant alterations of joint movement in patients compared to healthy control and explained how motions for related joints are correlated.

This manuscript is a well-written paper overall. However, providing more detailed descriptions would be beneficial for the better comprehension of readers.

Response : We sincerely appreciate your positive evaluation. We fully agree with your suggestion that providing more detailed descriptions would enhance the readers' understanding. In response, we have added additional details to improve the clarity of the manuscript.

Specific Comments 1: The descriptions of data acquisition are vague. Some explanations about TDPT-GT and data acquisition protocol may be found in section 1 and section 2.2. But it would be better to organize the description separately with a full explanation about it.

Response 1: Thank you for providing valuable and instructive comments and suggestions. We agree with this comment. Therefore,we made a separate section of data acquisition, 2.3. ‘Three‐Dimensional Pose Tracker for Gait Test (TDPT‐GT) app for 3D human motion estimation’ in the methods section.

Specific Comments 2: The 3D coordinate data with a confidence score of > 0.7 were used for analysis. What is the definition of confidence score and how they were calculated? Similarly, why the 75% tolerance ellipse were used to describe the distribution?

Response 2: Thank you for your instructive comment. In the first paper, confidence scores were referred to as AI scores. These scores represent the accuracy and certainty of the 3D coordinates estimated by deep learning, with values closer to 1 indicating higher certainty. Scores below 0.7 were deemed unreliable and not used. Additionally, both raw and smoothed coordinates were low-pass filtered, and the AI score, reflecting the certainty and probability of the coordinate estimation, was automatically calculated and saved as a CSV file on the iPhone. We added one sentence in the methods section.

 Additionally, we adopted the 75% confidence interval because 75% was determined based on the distribution of outliers.

Specific Comments 3: (Recommendation) The results showed general alternations of joint movements in patients compared to healthy control as group analysis. I think those features would have some relationship with quantitative gait indices or severity of mobility. How about performing subject-wise analysis to investigate those relationships?

Response 3: Thank you for your recommendation and for providing constructive feedback. The patient data used in this study did not include detailed medical histories for individual patients, including severity of mobility, making subject-wise analysis not feasible. Therefore, we limited our analysis to comparing overall trends between groups.

Reviewer 2 Report

Comments and Suggestions for Authors

Dear authors,

It was a pleasure to read your paper. I have one minor comment though.

In lines 96-97  you state "The gaits of the patients with HD, PD, and CM were evaluated using TDPT-GT during simple walking around a 1-m-diameter circle twice at several timings: for example,...".

Please add a statement justifying why you selected the particular walking path, naming a 1-m-diameter circle, supporting your justification with the appropriate references.

No other comment.

Author Response

Comments and Suggestions for Authors:

In lines 96-97  you state "The gaits of the patients with HD, PD, and CM were evaluated using TDPT-GT during simple walking around a 1-m-diameter circle twice at several timings: for example,...".

Please add a statement justifying why you selected the particular walking path, naming a 1-m-diameter circle, supporting your justification with the appropriate references.

No other comment.

Response : The selection of a 1-meter-diameter circular walking path was driven by the need to keep the participant’s full body—from head to toes—within the smartphone’s camera view while assessing walking. Straight-line paths required frequent adjustments of the smartphone, which made the participant appear smaller and potentially compromised the accuracy of the evaluation. After experimenting with various options, we determined that a 1-meter-diameter circle provided the optimal balance between maintaining a consistent camera view and facilitating accurate gait assessment. This approach was based on practical considerations rather than specific references from past literature.

Reviewer 3 Report

Comments and Suggestions for Authors

This paper quantified the characteristics of pathological limb movements in patients with Hakim's disease (HD), Parkinson's disease (PD), and cervical myelopathy (CM). The research can help reduce the development of these diseases. However, the following questions still need to be clarified or improved before the manuscript is published:

1.      What is the model of the instrument used to collect these data? It is recommended that the collection and processing of these data be described in detail in the manuscript.

2.      In general, the collected human motion data usually has the phenomenon of whole-body motion coupling. How was the independent motion data of each joint obtained in this manuscript?

3.      Are all the collected data for each joint distributed in the proposed elliptical area or the 3D space?

4.      Whether the ‘upper and lower limb’ in the title can be collectively called ‘limbs’.

5.      In the introduction, the new system proposed in line 58 of the manuscript is not introduced in detail. It is recommended to use a section in the main text with pictures to describe the configuration, input, and output of the device and how to obtain relevant data.

6.      Authors should explain the data mentioned below in detail? What is the unit of ‘0.1’on line 65 in page 2? In page 3, line 104, is the highest confidence value 1? Is it convenient to replace ‘0.7’ with ‘70%’, which is more suitable for confidence? What is the ‘1’ unit on line 120 in page 3?

7.      In Table 1, it is suggest that the authors divide the symptoms of the three diseases into mild, moderate, and severe respectively, and give the number of symptoms of each degree?

Author Response

Comments 1: What is the model of the instrument used to collect these data? It is recommended that the collection and processing of these data be described in detail in the manuscript.

Response 1: We sincerely thank you for your providing valuable and instructive comments and suggestions. For data collection, we used an iPhone SE2 purchased specifically for research purposes. The participants' gait was recorded as 2D video using this device. The data from the participants were anonymized before analysis.

Comments 2:    In general, the collected human motion data usually has the phenomenon of whole-body motion coupling. How was the independent motion data of each joint obtained in this manuscript?

Response 2: Thank you for pointing this out. In this application, to estimate the relative 3D coordinates of human motion, 24 key points are extracted using the modified ResNet34 and the 3D heatmap method, without estimating any connections between the points. The 3D heatmap method consists of 28 × 28 × 28 blocks for each of the 24 key points. The most likely (hottest) block for each key point is identified by the 3D heatmap and matched with the key points extracted from the previous two frames. Therefore, we obtain independent motion data for each joint without establishing connections between the joints. According to the reviewer's advice, we made a separate section of data acquisition, 2.3. ‘Three‐Dimensional Pose Tracker for Gait Test (TDPT‐GT) app for 3D human motion estimation’ in the methods section.

Comments 3: Are all the collected data for each joint distributed in the proposed elliptical area or the 3D space?

Response 3: We sincerely appreciate your valuable feedback. Only data with an AI confidence score of 0.7 or higher are retained in the 3D space. In addition, the elliptical region represents the 75% confidence area, with the remaining 25% considered as outliers. These outliers are assumed to be distributed within the 2D relative coordinates, rather than in 3D space.

Comments 4:  Whether the ‘upper and lower limb’ in the title can be collectively called ‘limbs’.

Response 4: We are grateful for your careful review and suggestions. As the main point of this paper includes the ability to assess upper limb movements in addition to the lower limb assessment in our previous study. Therefore, we would like to use the term 'upper and lower limbs' in the title.

Comments 5:  In the introduction, the new system proposed in line 58 of the manuscript is not introduced in detail. It is recommended to use a section in the main text with pictures to describe the configuration, input, and output of the device and how to obtain relevant data.

Response 5We truly appreciate your recommendation. According to the reviewer's advice, we made a separate section of data acquisition, 2.3. ‘Three‐Dimensional Pose Tracker for Gait Test (TDPT‐GT) app for 3D human motion estimation’ in the methods section. In addition, we have added one video (Supplementary File) to help you better understand TDPT-GT application, in response to a reviewer's request.

Comments 6:  Authors should explain the data mentioned below in detail? What is the unit of 0.1on line 65 in page 2? In page 3, line 104, is the highest confidence value 1? Is it convenient to replace 0.7 with 70%, which is more suitable for confidence? What is the 1 unit on line 120 in page 3?

Response 6: Thank you for your comments.

①    The values represent the ratio relative to the length of the lower limbs, which was set to 1 in our analysis of pathological gait using this application; therefore, the values have no units.

②    '0.7' is the confidence score automatically calculated by the deep learning algorithm, without units, and it cannot be converted into a percentage.

③    Similarly, there are no units because the values are ratios, assumed to be 1 when converting measurements to relative coordinates in the application.

Comments 7:  In Table 1, it is suggest that the authors divide the symptoms of the three diseases into mild, moderate, and severe respectively, and give the number of symptoms of each degree?

Response 7: We did not subjectively rate the severity of symptoms for each patient.

Round 2

Reviewer 3 Report

Comments and Suggestions for Authors

I have no further comments.